# Ultrasound-Guided OnabotulinumtoxinA Injections to Treat Oromandibular Dystonia in Cerebral Palsy

**DOI:** 10.3390/toxins14030158

**Published:** 2022-02-22

**Authors:** Fabiola I. Reyes, Hannah A. Shoval, Amy Tenaglia, Heakyung Kim

**Affiliations:** 1Scottish Rite for Children, Dallas, TX 75219, USA; 2Department of Physical Medicine and Rehabilitation, The University of Texas Southwestern Medical Center, Dallas, TX 75390, USA; 3Children’s Specialized Hospital, Clifton, NJ 07013, USA; aurashoval@gmail.com; 4Department of Physical Medicine and Rehabilitation, Rutgers New Jersey Medical School, Newark, NJ 07101, USA; 5Department of Rehabilitation and Regenerative Medicine, Columbia University Irving Medical Center, New York, NY 10032, USA; alt9109@nyp.org (A.T.); hk2641@cumc.columbia.edu (H.K.)

**Keywords:** oromandibular dystonia, lateral pterygoid, cerebral palsy, ultrasound guidance, botulinum toxin

## Abstract

Cerebral palsy (CP) is a group of non-progressive disorders of motor function in children resulting from an injury to an immature brain. In addition to abnormal limb and trunk movement, individuals with CP can experience involuntary muscle contractions of the lower facial muscle groups, causing oromandibular dystonia (OMD). Contraction of the lateral pterygoids and submandibular muscles depresses the mandible. OMD involving the lateral pterygoids can therefore lead to involuntary jaw opening posture, affecting the ability to feed and speak effectively. We present a case series of five patients with CP and OMD that received novel ultrasound-guided onabotulinumtoxinA to the lateral pterygoid muscles. Our goal was to determine if chemodenervation would improve the mouth-closing ability, thus in turn improving the ability to swallow, chew, manage secretions, and communicate. We describe this unique injection method and report a subjective improvement in eating abilities and communication, in addition to a positive upward trend in most patients’ weights, with no significant adverse side effects.

## 1. Introduction and Literature Review

Cerebral palsy (CP) is a group of non-progressive disorders of motor function resulting from an injury to an immature brain. CP is characterized by abnormal muscle tone, such as spasticity and dystonia, and can be associated with sensory, proprioceptive, and cognitive dysfunction [1]. Dystonia, which is characterized by involuntary muscle contractions causing muscle spasms and abnormal postures or positions, is particularly important to recognize and treat in CP. Importantly, it can cause pain and interfere with function. A recent study showed that dystonia, but not spasticity, was correlated with both the Gross Motor Functional Scale (GMFCS) and Manual Ability Classification System (MACS) [2]. Dystonia can be idiopathic, genetic or acquired/secondary, most commonly due to brain injury or medication usage. In the case of CP, dystonia is thought to be secondary due the brain injury.

Involuntary oropharyngeal and jaw muscle contractions that lead to intermittent or sustained postures in the jaw, mouth, and tongue, can be described by the term oromandibular dystonia (OMD). Abnormal jaw movements can be challenging as they can lead to trismus, bruxism, jaw pain, and impairment in communication, swallowing, chewing, and managing secretions [3,4,5]. Without proper identification and management, this can lead to growth and developmental delays and decreased social participation [6,7].

Oromandibular dystonias can be classified by the direction of jaw movement: jaw opening (lateral pterygoids and digastrics), jaw closing (masseters, temporalis, medial pterygoid), and jaw deviation (contralateral lateral pterygoids, ipsilateral temporalis) [8]. For jaw opening dystonia, in addition to the main muscles of lateral pterygoid and digastrics, the platysma, mylohyoid, and geniohyoid may also be involved [9].

Botulinum toxin is a biologic neurotoxin that inhibits exocytosis of acetylcholine in presynaptic nerve terminals, causing temporary functional denervation of muscles [10,11]. This paralytic effect has allowed the use of botulinum toxin in the treatment of motor disorders characterized by dystonia and spasticity [5]. Given the functional impairments due to these particular movement patterns in CP, botulinum toxin has thus become a common tool used to address spasticity and dystonia associated with CP [12,13].

We present a series of five patients with CP, aged one to 28 years, with focal OMD, jaw-opening type, secondary to CP involving primarily the lateral pterygoid (LPt) muscles who received ultrasound-guided onabotulinumtoxinA injections to these muscles. The LPt muscles consist of two heads, the inferior or lower head and the superior or upper head [14]. The primary role of the inferior head is mouth opening and if hypertonic, we propose, can contribute to mouth-closing dysfunction. The purpose of injecting the LPt muscles was to determine if chemodenervation would help improve mouth-closing ability measured by caretaker report. Patient-reported changes in functions associated with chewing, swallowing, growth, communication, and social interaction were also included in this case series. All patient visits that occurred between 25 June 2014 and 20 December 2016 were reviewed, and a total of five patients who received onabotulinumtoxinA injections to the LPt muscles were identified. Pre-injection evaluations, procedure notes, and follow-up clinic notes through 05/2017 were reviewed for indications, dosing, complications, results, and side effects noted by caretakers.

Prior to injections, patients were evaluated by a pediatric physiatrist in an outpatient clinic. The decision to inject the LPt muscles was determined based on report by patients and caregivers of difficulty approximating lips, as well as reports of functional deficits due to mouth position. The bilateral LPt muscles were chosen given they are the predominant muscles that open the jaw. LPt muscle injections were performed under sedation in a sterile manner. Patients were sedated due to patient difficulty with remaining still during the procedure in the setting of needing to inject the lateral pterygoid, which is a relatively small and deep muscle. Difficulty remaining still was due to a variety of factors including the dystonia itself, young age, anxiety regarding the procedure, as well as sensitivity to needles and pain. The head was rotated to the contralateral side of the intended muscle target. The ultrasound probe was placed horizontally over the pre-auricular area of the face just in front of the tragus. The space bordered by the zygomatic arch cranially, the condylar process posteriorly, and the coronoid process anteriorly was palpated and aligned with the center of the ultrasound probe (Figure 1a,b). The needle was inserted into the space in an out-of-plane position, and a step-down technique was used for continued visualization of the needle as it passed through the masseter muscles to reach the LPt muscles (Figure 1c). Of note, the parotid gland lies superficial and posterior to the masseter muscles and could be visualized first, especially if the probe is placed close to the tragus. Caution was taken to identify and avoid the maxillary and temporal arteries, which can course superficial to the LPt muscles. Needle placement into the LPt inferior head was confirmed with ultrasound image and electrical stimulation at 2–3 mA, which caused mouth opening. OnabotulinumtoxinA dosing per muscle ranged from 0.47 units/kg/LPt to 1.05 units/kg/LPt. The dosing was based on experience with similarly sized muscles. The decision to perform repeat injections was then determined at follow-up visits based on recurrence or continued functional deficits associated with the mouth-opening dystonia.

## 2. Cases

### 2.1. Case 1

P1 is a male with mixed dystonic/spastic quadriplegic CP, i.e., GMFCS-IV associated with uterine rupture at birth and MRI findings of hypoxic changes in the hippocampi. At 22 months, he was starting to vocalize monosyllabic sounds but had difficulty closing his mouth, creating a barrier to progress in speech therapy. His goals for LPt chemodenervation were improved lip approximation and participation in speech therapy.

P1 received three sets of injections between the ages of 23 and 31 months (Table 1). After the injections, his caretaker reported improved ability to maintain his mouth closed, better chewing quality, and fewer breaks taken while eating. At follow-up 26 months after the third set of injections, his caretaker reported continued positive effects, and further injections were not planned. No adverse effects were reported throughout the study period. Regarding his recorded weights, there was a decrease noted between the second and third set of injections, along with a decrease in height by 5 cm. We postulate that the measurements taken the day of the third set of injections may be an outlier or inaccurate given he did not lose height and all subsequently documented heights and weights demonstrated a positive trend.

### 2.2. Case 2

P2 is a female with dyskinetic quadriplegic CP, i.e., GMFCS-V related to a perinatal hypoxic event after premature birth. At nine years of age, she spoke only a few words, mostly “yes” and “no”. On exam, she had difficulty controlling facial movements with frequent tongue thrusting and had difficulty closing her mouth to chew and speak. Her goals for LPt chemodenervation were improved mouth closure and increased production speech.

P2 received nine sets of injections between the ages of nine and 11 years (Table 1). After the injections, her caretaker reported improved mouth-closing ability and quality of speech. After the 8th injection, her caretaker also noted decreased tongue thrust. At follow-up five months after the 9th injection, the plan was to repeat the LPt injections. No adverse effects were reported during the study period, and there was a positive trend in weight.

### 2.3. Case 3

P3 is a female with mild mixed dystonic/spastic quadriplegic CP, i.e., GMFCS-II and MRI findings of bilateral MCA territory infarcts. She was referred at six years of age for injections to the salivary glands for sialorrhea management. On exam, her mouth was maintained in an open position, leading to severe anterior drooling. Her speech was difficult to understand, and she used gestures and signs to communicate. Both LPt muscle and salivary gland injections were planned with the goals of improving mouth closure and decreasing anterior drooling.

P3 received four sets of injections between the ages of seven and eight years. At her follow-up visits, she was noted to have improved mouth closure and speech. P3′s mother also reported that she appeared less frustrated when communicating and that her behavior had improved. After the last set of injections, she started using a straw to drink thick liquids consistent with her mom’s report of improved oral intake. The plan was for repeat injections and continuing oromotor strengthening exercises in speech therapy. No adverse effects were reported during the study period, and there was a positive trend in weight.

### 2.4. Case 4

P4 is a female with mixed dystonic/spastic quadriplegic CP, i.e., GMFCS-IV related to premature birth at 28 weeks gestational age with imaging findings of periventricular leukomalacia. At four years of age, she was starting to put two words together but had difficulty with articulation due to the inability to fully close her mouth. She was eating a soft diet by mouth but required long periods of time to complete her meals. Her goals for LPt chemodenervation were to improve articulation and eating ability.

P4 received two sets of injections between the ages of four and five years (Table 1). After the first set of injections, her parents reported improvement in chewing with decreased time to complete meals. She was also more verbal, and her parents were able to understand her speech 100% of the time. After the second set of injections, her parents observed improved swallowing abilities and increased oral intake. She was also noted to have a positive weight trend after having lost weight in the 5-month period preceding her first injection.

### 2.5. Case 5

P5 is a female adult with mixed dystonic/spastic quadriplegic CP, i.e., GMFCS-V. She presented with a chief complaint of bruxism, in addition to difficulty closing her mouth. Injections to both the LPt muscles to facilitate jaw closing and the masseter and temporalis muscles to alleviate bruxism were planned, with the additional goal of improving jaw alignment.

At follow-up, two months after injections, her caretaker reported improved jaw alignment as her teeth were better approximated. She was eating better with improved ability to keep food in her mouth, and her bruxism and jaw pain were improved. She was noted to have a slight increase in drooling. The plan was to continue monitoring. No major adverse events were reported, and her weight increased after the injections.

## 3. Discussion

Untreated oromandibular dystonia (OMD) can lead to difficulty in opening and/or closing the mouth resulting in trismus, pain, impaired communication, and difficulty with chewing, swallowing, and managing secretions. It can also affect management during critical medical care, as trismus in people with CP can lead to significant difficulties with endotracheal tube placement during intubation. Botulinum toxin has been used to manage multiple manifestations of OMD, including trismus and bruxism, with success [14,15,16,17,18]. Recent studies have shown that botulinum toxin injections to dystonic LPt muscles can successfully treat mouth-closing difficulties in adults [9,19]. Currently, there is no reported literature on managing OMD involving primarily the LPt muscles in children and adults with CP. Our case series is the first to show successful management of OMD in this unique patient population and to examine particular outcomes, such as improvement in communication.

In our clinical experience, dystonia, in people with CP as well as other brain injuries, is often under recognized. Lumsden et al., found a frequency of 70% of patients with CP have a combination of spasticity and dystonia. It is likely under-recognized due to being overshadowed by spasticity, as well as due to variability in dystonia types and presentation and difficulty measuring them [2]. Another barrier is assuming that oromotor difficulty in a person with CP is purely due to weakness and to not look for other causes, such as dystonia. Furthermore, even when OMD is recognized in CP, a practitioner may not recommend onabotulinum toxin for fear of weakening oromotor muscles.

By facilitating mouth closure in patients with CP, there was a subjective improvement in eating abilities and communication as reported by patients and caregivers. As there was a positive uptrend in most of the patients’ weights, further investigation is needed on how injections to LPt muscles and other jaw muscles affect nutrition, growth, and development. It is reasonable to postulate that by controlling movement not only in limbs but also in the jaw and facial muscles, better nutritional status may be achieved. This is especially critical in the CP population, as spasticity can cause an increased resting metabolic rate requiring increased nutritional intake

Current literature suggests that botulinum toxin has a good safety profile for OMD [17,18]. Consistent with this research, we did not observe any major adverse effects after the LPt muscle injections in our case series. P5, a 28-year-old female, did note mildly increased anterior drooling after injections, but this did not impair her ability to eat or swallow. This could have occurred given she intentionally received botulinum toxin into both her jaw-opening muscles (LPt) as well as jaw-closing muscles (masseter and temporalis). In an attempt to mitigate this, dosing to the masseter could be decreased to avoid significant weakening of the jaw-closing muscles.

Adverse events after botulinum toxin injections are typically caused by inaccuracies in dosing and/or injection techniques. Appropriate dosing is particularly important to minimize the risk of local spread of botulinum toxin. Rare but serious side effects that have been reported, such as difficulty swallowing or respiratory depression, have been attributed to the diffusion of the medication into distant muscles [20]. Dosing used in this study ranged from 10 to 20 units per muscle (0.47–1.05 units/kg/LPt) and was safe and effective. This dose is considerably smaller than the dosing reported in the adult literature [17]. Our recommendation in pediatric patients is to use the smaller dosages as used in this study; however, the minimal side effects noted in adults despite the large doses of toxin indicates that there is likely a large safety window [17].

In addition to appropriate dosing, accurate technique is important to prevent the inadvertent injection of botulinum toxin into an unintended muscle of mastication. The use of ultrasound guidance for LPt injections has not been reported in the literature to treat OMD in CP; however, it has been studied in masseter injections to treat bruxism [21]. Several systematic reviews have demonstrated that instrument-guided injections have a level 1 accuracy [22,23]. Injections to treat bruxism performed with ultrasound guidance were more accurate than with the use of anatomical landmarks alone [21]. With anatomical guidance only, large muscles have an accuracy of >75%, and small muscles have an accuracy as low as 11% [24,25]. The LPt muscles are deep and unable to be palpated; therefore, it is essential to use ultrasound and/or electrical stimulation for accurate localization. All the procedures for the five cases were done under instrument guidance with both ultrasound and electrical stimulation. This precise injection technique could be another reason for positive outcomes after injections without significant adverse events.

There were a few limitations of this study. Given this was a retrospective chart review, the dosing between patients was not consistent, and the outcome measurements for communication, swallowing, chewing, drooling, and pain were all subjective. Additionally, a small number of patients were reviewed. To better demonstrate the efficacy and safety of LPt muscle injections with botulinum toxin, prospective and controlled studies with more patients and clearly delineated goals with standardized outcome measures are needed.

In conclusion, injection of onabotulinumtoxinA into the LPt muscles of patients with OMD involving primarily the LPt muscles appears to be safe and effective. This could have a significant impact on patients with CP by improving their ability to swallow, chew, manage secretions, and communicate. This in turn can improve social and emotional well-being, leading to an overall improvement in quality of life.

## Figures and Tables

**Figure 1 toxins-14-00158-f001:**
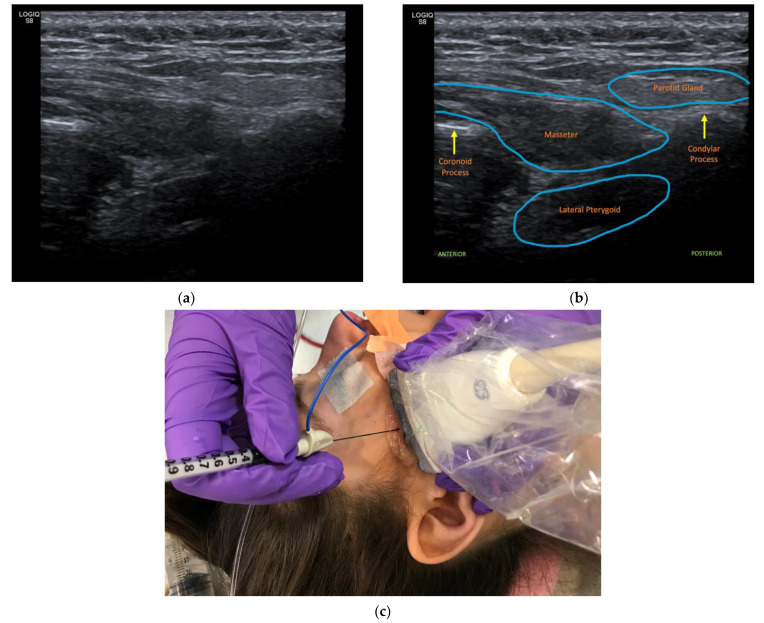
(**a**) Ultrasound view without label, (**b**) Ultrasound view with muscle and bone labeled, (**c**) Placement of ultrasound probe and needle.

**Table 1 toxins-14-00158-t001:** Summary of patient characteristics, dosing, results, and weight trends.

Patient Information	Baseline	Injections (Injection Number: Age, Total Units/LPt, (Units/kg/LPt))	Results as Observed/Reported by Care Takers and Physician	Weight Trends (Injection Number: Weight)
Patient 1 Dx: mixed spastic/dystonic quadriplegic CP GMFCS: IV	General: Mouth open 75% of the time. Feeding: difficulty inserting spoon into mouth when feeding Speech: vocalizing monosyllabic sounds such as “ah, ah” Quality of Life: difficulty working with speech therapist	#1: 1 y 11 mo; 5 units /LPt (0.47 units/kg/LPt) #2: 2 y 2 mo; 10 units/ LPt (0.84 units/kg/LPt) #3: 3 y 7 mo; 10 units/ LPt (0.87 units/kg/LPt)	#1: Mouth open 40% of time, improved ease of feeding/clearing spoon when inserting into mouth #2: improved ability to close mouth, able to chew food and eat without breaks, ability to close mouth further improved by 50% #3: continued good mouth closure, improved speech, starting to close lips to swallow	#1: 10.4 kg #2: 11.6 kg #3: 10.5 kg Last Documented Age: 4 y 9 mo Weight: 16 kg
Patient 2 Dx: dyskinetic quadriplegic CP GMFCS: V	General: inability to control facial muscles, difficulty closing mouth, mouth remains open 80% of time but able to close mouth with masseter stimulation Feeding: difficulty chewing food Speech: has a few words, difficulty speaking Quality of Life: difficulty expressing emotions	#1: 9 y 0 mo; 10 units /LPt (0.625 units/kg/LPt) #2–7: 9 y 4 mo–11 y 1 mo; 10–15 units /LPt (0.63–0.75 units/kg/LPt) * #8: 11 y 6mo; 20 units/ LPt (1 unit/kg/LPt) #9: 11 y 10 mo; 20 units/LPt (0.99 units/kg/LPt)	# 1: starting to talk more, mouth appears more relaxed and can now close mouth volitionally # 2–8: improved ability to close mouth, improved quality and quantity of speech, takes less time to eat, mouth open 20% of time with further (60%) improvement after more injections, decreased tongue thrust from a severity of 9/10 to 2/10 on a Likert scale #9: difficulty closing mouth after longer than usual interval between injections (5 months), jaw is noted to be open throughout clinic visit	#1 16.1 kg #2–7: 15.85–19.95 kg #8: 19.5 kg #9: 20.3 kg Last Documented Age: 12 y 3 mo Weight: 38kg
Patient 3 Dx: mixed spastic/dystonic quadriplegic CP GMFCS: II	General: able to close mouth on command, but maintains it open for most of the exam Feeding: chokes and coughs when swallowing liquids while sitting Speech: difficult to understand, relies on gestures and signs to communicate Quality of Life: excessive drooling, saliva on clothes with bib change 15 times per day	#1: 7 y 2 mo; 20 units/ LPt (1.05 units/kg/LPt) #2: 7 y 5mo; 20 units/LPt (1.01 units/kg/LPt) #3: 7 y 9 mo; 20 units/LPt (0.96 units/kg/LPt) #4: 8 y 2 mo; 20/units/ LPt (0.9 units/kg/LPt)	#1: improved ability to close mouth and speak, appeared less frustrated with better behavior, bib change 5 times per day #2-3: improved ability to close mouth and speak #4: starting to use straw to drink thick liquids, injection effect lasting longer, and improved oral intake	#1: 18.9 kg #2: 19.7 kg #3: 20.9 kg #4: 22 kg Last Documented As per #4
Patient 4 Dx: mixed spastic/dystonic quadriplegic CP GMFCS: IV	General: poor motor control of the mouth Feeding: eats soft diet and takes longer than typical to finish meals Speech: can speak 1–2 words together but difficult to articulate Quality of Life: excessive drooling	#1: 4 y 7 mo; 10 units/ LPt (0.63 units/kg/LPt) #2: 5 y 0 mo; 10 units/LPt (0.6 units/kg/LPt)	#1: improved chewing and overall eating, improved articulation, parents can understand speech 100% of the time and strangers can understand speech 60% of the time #2: improvement in chewing, swallowing, and amount of PO intake	#1: 14.1 kg #2: 14.6 kg Last Documented Age: 6 y 5 mo Weight: 16.3 kg
Patient 5 Dx: mixed spastic/dystonic quadriplegic CP GMFCS: V	General: frequent bruxism and mal-alignment of the jaw Feeding: difficulty with chewing and eating Quality of Life: pain with teeth grinding	#1 28 y/o; 20 units/LPt (0.44 units/kg/LPt)	#1: improved jaw alignment, improved ability to chew and eat, improved teeth approximation, able to keep food in mouth while eating, slight increase in drooling, pain resolved	#1: 45.4 kg Last Documented Age: 28 y/o., (3 months post injections) Weight: 45.81

* Time interval between injections ranged from three to five months; LPt = Lateral Pterygoid.

## Data Availability

Not applicable.

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
