# Peer review of "Ultrasound-Guided OnabotulinumtoxinA Injections to Treat Oromandibular Dystonia in Cerebral Palsy"

_toxins, 2022, doi:10.3390/toxins14030158_

Round 1
Reviewer 1 Report
Critical review
Abstract:
Line 5 “in addition to difficulty”, what does difficulty imply- please clarify
Line 6 “facial muscle dysfunction”- what does dysfunction imply- please clarify
Lines 5-11: Muscle action is inaccurate/not detailed. Lateral pterygoid and submental muscle complex cause jaw opening. Medial pterygoid, masseter and temporalis causes jaw closure. Please provide accurate anatomical function and associated references.
Introduction:
Lines 28-31: The line there is too generic “facial muscles are affected”, affected how. Please provide reference where oromandibular dystonia can lead to trismus. The 2 references provided do not mention trismus.
Line 33- Please specify what type of dystonia your patients had here.
Line 50- Why was it necessary to sedate the patients- please mention the reason.
Please give a couple of line description about what dystonia is, what are the types of oromandibular dystonia with muscles affected for each type.
Discussion:
Line 151- Numerous articles talks about efficacy of botulinum toxin in oromandibular dystonia and ultrasound guidance. Alter K in Seminal of Neurology 2016 is a good review on this topic. Authors mention that, what is unique about this article is that this is in patients with cerebral palsy. How is this patient population unique with regards to their dystonia?
Line 169- appropriate dosing reduces risk of systemic spread. Please provide references of articles confirming systemic spread. If not please change from “systemic spread” to “local spread”.
Author Response
Abstract:
Line 5 “in addition to difficulty”, what does difficulty imply- please clarify
Thank you for the comment and for pointing out that the phrase is too general We have changed the wording to be clearer. Please see lines 5-7, which include the following changes:
Original: In addition to difficulties with limb and trunk movement, children with CP can experience facial muscle dysfunction, termed oromandibular dystonia (OMD)
Changes: In addition to abnormal limb and trunk movement, individuals with CP can experience involuntary muscle contractions of the lower facial muscle groups, causing oromandibular dystonia (OMD)
Line 6 “facial muscle dysfunction”- what does dysfunction imply- please clarify
Thank you for asking for clarification. By using the definition provided in 2020 review on Blepharospasm and OMD in Toxin by Hassell and Charles, we have clarified this point. Please see lines 6-7 in the revised manuscript, which includes the wording below:
Original: In addition to difficulties with limb and trunk movement, children with CP can experience facial muscle dysfunction, termed oromandibular dystonia (OMD)
Changes: In addition to abnormal limb and trunk movement, individuals with CP can experience involuntary muscle contractions of the lower facial muscle groups, causing oromandibular dystonia (OMD)
Lines 5-11: Muscle action is inaccurate/not detailed. Lateral pterygoid and submental muscle complex cause jaw opening. Medial pterygoid, masseter and temporalis causes jaw closure. Please provide accurate anatomical function and associated references.
We agree that the description of oromotor movements should be more detailed. We have included a more detailed description of the muscles that aide in jaw opening and jaw closing functions. Please see lines 7-10 in the revised manuscript, which includes the wording below:
Original: One cause could be spastic or dystonic lateral pterygoid muscles, which are the main muscles responsible for opening the mouth. OMD involving these muscles can lead to difficulty with closing the mouth, affecting the ability feed and speak effectively.
Changes: Contraction of the lateral pterygoids and submandibular muscles depress the mandible. OMD involving the lateral pterygoids can therefore lead to involuntary jaw opening posture, affecting the ability feed and speak effectively.
Introduction:
Lines 28-31: The line there is too generic “facial muscles are affected”, affected how. Please provide reference where oromandibular dystonia can lead to trismus. The 2 references provided do not mention trismus.
Thank you for the comment. We agree that this term is also too generic, and so we have used new references to clarify. We have also added a reference that also addressed trismus and bruxism. The following changes were made, which can be found on lines 36-41 in the document:
Original:
When facial muscles are affected, termed oromandibular dystonia (OMD), opening and/or closing the mouth can be challenging leading to trismus, jaw pain, and impairment in communicating, swallowing, chewing, and managing secretions. Without proper identification and management, this can lead to growth and development delay, in addition to decreased social participation
Changes:
Involuntary oropharyngeal and jaw muscle contractions that lead to intermittent or sustained postures in the jaw, mouth and tongue, can be described by the term oromandibular dystonia (OMD). Abnormal jaw movements can be challenging as they can lead to trismus, bruxism, jaw pain, and impairment in communication, swallowing, chewing, and managing secretions [3-5]. Without proper identification and management, this can lead to growth and developmental delays and decreased social participation [6,7].
Line 33- Please specify what type of dystonia your patients had here
Thank you, we have added more detail to specify the type of dystonia. Please seen lines 53-54 in the edited manuscript.
Original:
We present a series of five patients with CP ages one to 28 years with OMD involving primarily the lateral pterygoid (LPt) muscles who received ultrasound-guided onabotulinumtoxinA injections to these muscles
Changes:
We present a series of five patients with CP ages one to 28 years with focal OMD, jaw opening type, secondary to CP involving primarily the lateral pterygoid (LPt) muscles who received ultrasound-guided onabotulinumtoxinA injections to these muscles.
Line 50- Why was it necessary to sedate the patients- please mention the reason.
Thank you for the question. We have explained why the patients were sedated. Please see lines 72-76 for the added explanation.
Changes:
Patients were sedated due to patient difficulty with remaining still during the procedure in the setting of needing to inject the lateral pterygoid which is a relatively small and deep muscle. Difficulty remaining still was due to a variety of factors including the dystonia itself, young age, anxiety regarding the procedure, as well as sensitivity to needles and pain
Please give a couple of line description about what dystonia is, what are the types of oromandibular dystonia with muscles affected for each type.
Thank you for the feedback. We have added descriptions of dystonia in general, and the types of oromandibular dystonia. Please see lines 28-30 and 42-46 in the edited manuscript
Changes:
Dystonia, which is characterized by involuntary muscle contractions causing muscle spasms and abnormal postures or positions, is particularly important to recognize and treat in CP
Oromandibular dystonias can be classified by the direction of jaw movement: jaw opening (lateral pterygoids and digastrics), jaw closing (masseters, temporalis, medial pterygoid) and jaw deviation (contralateral lateral pterygoids, ipsilateral temporalis) [8]. For jaw opening dystonia, in addition to the main muscles of lateral pterygoid and digastrics, the platysma, mylohyoid and geniohyoid may also be involved [9].
Line 151- Numerous articles talks about efficacy of botulinum toxin in oromandibular dystonia and ultrasound guidance. Alter K in Seminal of Neurology 2016 is a good review on this topic. Authors mention that, what is unique about this article is that this is in patients with cerebral palsy. How is this patient population unique with regards to their dystonia?
Thank you for this recommendation. We have added to both the introduction and to the discussion in order to highlight what is unique to patients with CP in regards to their Dystonia. Please see lines 31-35, 50-52 and 179-186 and 193-194 in the revised manuscript
Changes:
A recent study showed that dystonia, but not spasticity, was correlated with both the Gross Motor Functional Scale (GMFCS) and Manual Ability Classification System (MACS) [2]. Dystonia can be idiopathic, genetic, or acquired/secondary, most commonly due to brain injury or medication usage. In the case of CP, dystonia is thought to be secondary due the brain injury.
In our clinical experience, dystonia, in people with CP as well as other brain injuries is often under recognized. Lumsden et al., found a frequency of 70% of patients with CP have a combination of spasticity and dystonia. It is likely under-recognized due to being overshadowed by spasticity, as well as due to variability in dystonia types and presentation and difficulty measuring them [2]. Another barrier is assuming that assuming that oromotor difficulty in a person with CP is purely due to weakness and not look for other causes, such as dystonia. Furthermore, even when OMD is recognized in CP, a practitioner may not recommend onabotulinum toxin for fear of weakening oromotor muscles.
This is especially critical in the CP population as spasticity can cause an increased resting metabolic rate requiring increased nutritional intake
Line 169- appropriate dosing reduces risk of systemic spread. Please provide references of articles confirming systemic spread. If not please change from “systemic spread” to “local spread”.
Thank you for catching this. It has been changed. Please see line 205 on the attached manuscript
Original: Appropriate dosing is particularly important to minimize the risk of systemic spread of botulinum toxin
Changes: Appropriate dosing is particularly important to minimize the risk of local spread of botulinum toxin

Reviewer 2 Report
The authors used ultrasounds to guide botulinum neurotoxin in the Lateral Pterygoid muscle. Although ultrasounds have been already used for bont injection in other conditions, here the novelty is the treatment of
children and adults with oromandibular dystonia as a consequence of cerebral palsy.
As indicated by the same authors, a major limitation of the study is its fully retrospective nature. Accordingly, there was not an appropriate study design (the dose and the time changed from one patient to another and in one case, p5, she received other muscle injections) as well as in the clinical evaluation of the patients. Also, the evaluation of the toxin effect was not standardized and relies on subjective assessment by patient and caregivers for a very little number of patients.
At the same time, the benefit of guiding bont injection with ultrasounds in deep muscles is clear, and the study can be considered as an important proof of principle for the extension of this technique to neurological conditions with similar hurdles. In addition, the study also clearly indicates that ultrasounds allows to inject botulinum toxin with very high accuracy while avoiding toxin diffusion to nearby muscles. This is important as it suggest that the dose could be escalated to have longer lasting effects, which would be a major goal in young patients with CP.
My only suggestion is to expand a bit the introdcution adding some background on Botulinum neurotoxin mechanism of action and as to how patients with CP can benefit from its use. This would be important to better present the rationale of the study.
Author Response
Reviewer 2
My only suggestion is to expand a bit the introdcution adding some background on Botulinum neurotoxin mechanism of action and as to how patients with CP can benefit from its use. This would be important to better present the rationale of the study.
Thank you for the suggestion. We have added this background information on botulinum neurotoxin and its benefits specifically to individuals with CP. Please see lines 31-35, 47-52 and 179-186 in the revised manuscript
Changes:
A recent study showed that dystonia, but not spasticity, was correlated with both the Gross Motor Functional Scale (GMFCS) and Manual Ability Classification System (MACS) [2]. Dystonia can be idiopathic, genetic, or acquired/secondary, most commonly due to brain injury or medication usage. In the case of CP, dystonia is thought to be secondary due the brain injury
Botulinum toxin is a biologic neurotoxin that inhibits exocytosis of acetylcholine in presynaptic nerve terminals, causing temporary functional denervation of muscles [10,11]. This paralytic effect has allowed the use of botulinum toxin in the treatment of motor disorders characterized by dystonia and spasticity [5]. Given the functional impairments due to these particular movement patterns in CP, botulinum toxin has thus become a common tool used to address spasticity and dystonia associated with CP [12,13].
In our clinical experience, dystonia, in people with CP as well as other brain injuries is often under recognized. Lumsden et al., found a frequency of 70% of patients with CP have a combination of spasticity and dystonia. It is likely under-recognized due to being overshadowed by spasticity, as well as due to variability in dystonia types and presentation and difficulty measuring them [2]. Another barrier is assuming that assuming that oromotor difficulty in a person with CP is purely due to weakness and not look for other causes, such as dystonia. Furthermore, even when OMD is recognized in CP, a practitioner may not recommend onabotulinum toxin for fear of weakening oromotor muscles.
